# Systematically and Comprehensively Understanding the Regulation of Cotton Fiber Initiation: A Review

**DOI:** 10.3390/plants12213771

**Published:** 2023-11-04

**Authors:** Zeyang Zhai, Kaixin Zhang, Yao Fang, Yujie Yang, Xu Cao, Li Liu, Yue Tian

**Affiliations:** 1College of Biotechnology, Jiangsu University of Science and Technology, Zhenjiang 212003, China; zhai98doc@163.com (Z.Z.); zhangkaixin202202@163.com (K.Z.); fangyao5627@163.com (Y.F.); yang1344813698@163.com (Y.Y.); caoxv618@vip.163.com (X.C.); touchliu@163.com (L.L.); 2Key Laboratory of Silkworm and Mulberry Genetic Improvement, Ministry of Agricultural and Rural Areas, Sericultural Research Institute, Chinese Academy of Agricultural Sciences, Zhenjiang 212018, China

**Keywords:** cotton, fiber initiation, molecular mechanism, phytohormones, transcription factors

## Abstract

Cotton fibers provide an important source of raw materials for the textile industry worldwide. Cotton fiber is a kind of single cell that differentiates from the epidermis of the ovule and provides a perfect research model for the differentiation and elongation of plant cells. Cotton fiber initiation is the first stage throughout the entire developmental process. The number of fiber cell initials on the seed ovule epidermis decides the final fiber yield. Thus, it is of great significance to clarify the mechanism underlying cotton fiber initiation. Fiber cell initiation is controlled by complex and interrelated regulatory networks. Plant phytohormones, transcription factors, sugar signals, small signal molecules, functional genes, non-coding RNAs, and histone modification play important roles during this process. Here, we not only summarize the different kinds of factors involved in fiber cell initiation but also discuss the mechanisms of these factors that act together to regulate cotton fiber initiation. Our aim is to synthesize a systematic and comprehensive review of different factors during fiber initiation that will provide the basics for further illustrating these mechanisms and offer theoretical guidance for improving fiber yield in future molecular breeding work.

## 1. Introduction

Cotton is an important economic crop worldwide. Cotton has four kinds of cultivated species, including *Gossypium Hirsutum*, *G. Barbadense*, *G. Arboreum*, and *G. Herbaceum*, of which the first two are allotetraploids (2*n* = 52) and the latter two are diploids (2*n* = 26) [1,2]. Fiber, a highly elongated and thickened single cell of the seed trichomes, is the most valuable product from cotton. Cotton fiber development is a very complicated process consisting of four continuous but overlapped stages: cell initiation (3 days before anthesis to 3 days post-anthesis (DPA), −3–3 DPA), cell elongation (3–20 DPA), cell wall thickening (20–45 DPA), and maturation (45–50 DPA) [3]. The number of fiber cell initials determines the cotton fiber yield, whereas cell elongation and cell wall thickening decide the final fiber quality. Upland cotton (*G. Hirsutum*) with high production and better yield potential is predominantly used as the commercial source of cotton fiber due to its wide adaptability to the environment [4,5]. Sea Island cotton (*G. Barbadense*) is famous for its long, strong, and fine fiber quality [4,5]. Traditional breeding methods have achieved great success in improving crop traits. However, due to the constraint of narrow genetic background in breeding materials, low efficiency of selection, and other multiple factors, the breeding of important crop varieties has entered a stage of gradual development in recent years. Therefore, accelerating innovation in breeding technology is imperative. An improvement in fiber yield and quality is always the major goal for cotton breeders; to fulfill the requirements of the modern textile industry, traditional breeding methods for cotton fiber yield and quality traits have been explored in different kind of populations, accompanied by cotton industrial development [6]. However, it is a huge challenge to improve fiber yield and quality simultaneously in cotton production.

Cotton fiber yield is essentially determined by the number of fiber cell initials on the seed ovule epidermis. Although cotton fiber initiation has been explored in the past, the molecular mechanism has not yet been well elucidated. Sequencing technology (such as genome sequencing, transcriptomes, and proteome sequencing) in cotton provided new tools with which to uncover the biological mechanisms underlying cotton fiber initiation [1,2]. The identification and characterization of genes contributing to the process are essential for understanding the biological roles and genetic interactions underlying fiber initiation. In addition, functional loci and molecular markers serve as starting points to improve fiber yield through the application of biotechnology, such as gene editing. These new approaches can be used by breeders to develop new varieties that improve cotton fiber yield, which would greatly contribute to cotton production.

Previous studies have shown that transcription factors (TFs) and phytohormones play important roles during cotton fiber development [7,8,9]. In this review, we mainly focus on studies related to cotton fiber initiation. We comprehensively summarize the inheritance and related candidate genes of cotton fiber initiation that have been identified in previous studies. We try to further clarify the regulatory relationships between these candidate genes and aim to provide a complete overview of the regulatory network underlying cotton fiber initiation.

## 2. Significance of Cotton Fiber Initiation

The initiation of cotton fiber is first visualized as the cell that protrudes from certain protodermal cells in cotton ovules before or on the day of anthesis [10]. The initiation process of fiber is rapid, which makes it hard to observe how protodermal cells differentiate into fiber cells. Biochemical analysis indicates that fiber cell fate determination appears before the formation of visible fiber initials [11]. Cotton fiber is a kind of single-cell trichome that differentiates from the epidermis of the ovule; the final fiber yield is determined by the number of epidermal cells that could potentially develop into fiber cells during the fiber initiation stage. Even though all epidermal cells of the cotton ovule have the potential to develop into fibers, only 25~30% of them ultimately become fibers [3]; this has become the restriction to improving cotton fiber yield. Therefore, it is possible to improve fiber yield by increasing the proportion of epidermal cells that could differentiate into fibers. Moreover, cotton fiber is the longest single cell in plants and could provide the ideal model system for studying diverse aspects of plant cell growth, including cell differentiation and elongation [12]. Thus, molecular investigations of fiber initiation will provide a theoretical basis for improving fiber yield, which is the main objective for breeders in the present and future work. Moreover, they will also enrich investigations of cell fate determination in plants.

Cotton fibers can be classified into two types: spinnable lint fiber and adherent fuzz fiber. Both are unicellular and tubular outgrowths that are indistinguishable in appearance during the early stages of fiber development, suggesting that their differentiation may involve a similar process and share a common pathway of initiation [13]. Lint, which is economically important, initiates on or before the day of anthesis (−1–0 DPA). Lint fibers grow to 2.5–3.5 cm in length, while fuzz fibers initiate on 4–5 DPA and ultimately reach approximately 5 mm in length [14]. Fuzz is found in most cotton varieties and aids the spreading of mature seeds by attaching to the fur of passing animals. However, fuzz is too short to be collected from the seeds through ginning, and any remaining fuzz on the seeds will enhance the cost of seed treatment for the following season [15]. While most cultivars developed in *G. Barbadense* are fuzzless, almost all cultivars in high-yield *G. Hirsutum* are fuzzy since fuzzless seeds are usually associated with a lower lint percentage, an important cotton yield component. Moreover, the ultimate capacity of cotton fibers is determined from the number of spinnable lint fibers. Cotton breeders have long aimed to cultivate cotton varieties with high lint percentage and fuzzless seeds to utilize their ginning advantages. To realize this target, it is necessary to uncover the mechanisms underlying the regulation of fiber initiation and to comprehend the biological function and genetic interactions.

## 3. Progress in Investigation of Cotton Spontaneous Initial Development Mutants

Spontaneous fiber mutants that inhibit the development of fiber initials provide powerful resources for studying the mechanism underlying the early stages of fiber development. Previous studies have described many cotton fiber mutants, including fuzzless, lintless, and fiberless (fuzzless–lintless) mutants. Their qualitative traits have been identified and described as shown in Table 1. The phenotype of representative fiber mutants is shown in Figure 1.

**Table 1 plants-12-03771-t001:** Different kinds of cotton fiber initiation-related mutants.

Phenotype	Variety	Genotype	Reference(s)	Resource
Fuzzless (no fuzz fibers)	Naked seeds 1 (N_1_NSM)	*N_1_N_1_*	[15,16,17,18]	USDA-ARS, College Station, TX, USA
	Naked seeds 2 (n_2_NSM)	*n_2_n_2_*	[16,17,18,19]	USDA-ARS, College Station, TX, USA
	Naked seeds 3 (n_3_)	*n_3_n_3_*	[20]	USDA-ARS, College Station, TX, USA
	Naked tufted seed (n_4_^t^)	*n_4_^t^n_4_^t^*	[21,22]	USDA-ARS, College Station, TX, USA
	GA0149	*FZ*	[23]	Institute of Cotton Research, Chinese Academy of Agricultural Sciences, Anyang, China
Lintless (shortened lint fibers <5 mm)	Ligon-lintless 1 (Li_1_)	*Li_1_Li_1_*	[24,25,26]	USDA-ARS, College Station, TX, USA
	Ligon-lintless 2 (Li_2_)	*Li_2_Li_2_*	[27,28,29]	USDA-ARS, College Station, TX, USA
	Ligon lintless-like mutant (Lix)	*LixLix*	[30]	Nanjing Agricultural University in China, Nanjing, China
Fiberless (no lint and fuzz fibers)	Xuzhou 142 fiberless mutant (XZ142FLM)	*n_1_n_1_n_2_n_2_li_3_li_3_*	[31]	Xuzhou Research Institute of Agricultural Science in China, Xuzhou, China
	SL1-7-1	*N_1_N_1_fl_1_fl_1_n_3_n_3_*	[32]	USDA-ARS, Stoneville, MS, USA
	MD17	*N_1_N_1_n_2_n_2_n_3_n_3_*	[20]	USDA-ARS, Stoneville, MS, USA

Two of the most well-described fuzzless mutants are the dominant *N_1_N_1_* and recessive *n_2_n_2_* loci, both producing naked seeds after ginning [15,16,17,18]. Homozygous *N_1_NSM* mutants are completely fuzzless, with a significantly reduced lint percentage [16,17,18]. Genetic mapping of the *N_1_N_1_* locus found that it is anchored to chromosome A12 and identified as a *MYBMIXTA*-like (*MML*) gene *GhMML3_A12* (or *GhMYB25*-like_*At*) [15]. A recent study further proposed that *MML3_D12* (or *MYB25*-like_*Dt*) is likely a major contributing locus for the recessive fuzzless trait (*n_2_n_2_*) in *G. Barbadense* [19]. In another study, another fuzzless seed loci (*n_3_*) was found to influence the effects of the *N_1_n_1_*, *N_2_n_2_*, and *N_3_n_3_* genes in lint percentage [20]. A newly found ethyl methanesulfonate-induced mutant *n_4_^t^* appears to be different from the above two naked seed loci; the mutation was shown to have a less severe negative influence on lint development [21,22]. Recently, a ~6.2 kb insertion larlNDEL_FZ_ was found to be related to fuzzless seeds and decreased trichomes in the *G. Arboreum* fuzzless mutant GA0149 in a genome-wide association study [23]. This insertion was proposed to enhance a dominant-repressor, *GaFZ*.

Although Ligon lintless-1 (*Li_1_*) and Ligon lintless-2 (*Li_2_*)are named as lintless mutants they are actually short fiber mutants. *Li_1_* is a dominant and monogenic mutant characterized by its short fibers (~5 mm in length) and the twisted growth of stems, leaves, and flowers in plants [24,25,26], while *Li_2_* exhibits similar fiber length but normal vegetative growth [27,28]. Using map-based cloning strategy, the *Li_1_* locus was identified as a single Gly65Val amino acid replacement in the actin polymerization domain of GhACT_LI1 (Gh_D04G0865) [26]. Recent studies revealed that the small interfering RNA (siRNA)-induced silencing of *RanBP1s* inhibits cotton fiber elongation in the *Li_2_* mutant [29]. *Lix* is a lintless loci and produces a plant morphology similarly to that of *Li_1_*; it is located on chromosome 4, which is homologous to chromosome 22 [30].

Some fiberless mutants have been described in *G. Hirsutum*, which include the Xuzhou 142 fiberless line (XZ142FLM) [31], SL1-7-1 [32], and MD17 [20]. However, the genotypes of these lines and their relationships still need to be clearly explained. Among these lines, XZ142FLM is the most studied fiberless mutant; it was originally discovered in a commercial upland cultivar, Xuzhou 142 (XZ142WT). So far, there are three kinds of genotypes for XZ142FLM: a three-locus type (*n_1_n_1_n_2_n_2_li_3_li_3_*) [31] and two four-locus types (*n_1_n_1_n_2_n_2_li_3_li_3_li_4_li_4_* and *n_1_n_1_n_2_n_2_li_3_li_3_n_3_n_3_*) [16,32]. The industrially important lint fiber-related gene (*Li_3_*) has been isolated and associated with another *MML* factor *GhMML4_D12* via a map-based cloning method [33]. Since some variations have been examined in many undescribed cotton fiber-related mutants [32], we believe that other genes which regulate fuzz or lint fiber initiations, besides these major loci, may exist. Moreover, no fuzzy and lintless phenotype has ever been discovered, indicating that fuzz genesbe epistatic to lint genes [16]. MD17 (*N_1_N_1_n_2_n_2_n_3_n_3_*) originated from the hybridization between dominant (*N_1_N_1_N_2_N_2_n_3_n_3_*) and recessive (*n_1_n_1_n_2_n_2_n_3_n_3_*) naked seed materials, suggesting that the interaction of fuzzless locus (*N_1_N_1_*, *n_2_n_2_*, and *n_3_n_3_*) could also result in the fiberless phenotype [20].

These mutants are significant genetic resources for identifying the genes related to fiber development. The genetic analysis and molecular mapping of these mutants are important steps in isolating these genes, and they may also offer additional clues to their organization and function.

## 4. Hormonal Control of Fiber Initiation and Development

Phytohormones, including auxin, gibberellin (GA), jasmonic acid (JA), ethylene (Eth), cytokinin (CK), abscisic acid (ABA), brassinosteroid (BR), salicylic acid (SA), and strigolactone (SL), are small endogenous signaling molecules in plants. Some of genes related to these hormones have been reported to regulate fiber development (Table 2).

### 4.1. Auxin

Auxin and gibberellin have well-elucidated roles in fiber cell initiation. For example, the addition of exogenous IAA and GA_3_ is indispensable for in vitro culturing of cotton unfertilized ovules [34]. Auxin plays important roles in numerous developmental process such as embryogenesis, root growth, and response to internal and external stimuli [35,36,37]. The accumulation of auxin is mainly from −5 to 10 DPA in the ovule epidermis and fiber cells, reaching its peak at 2–3 DPA [38]. Enhancing auxin levels at the appropriate time and locations during ovule and fiber development could increase cotton fiber yield [38]. Specific expression of auxin biosynthesis-related gene *iaaM* in cotton epidermal cells using the promoter of the petunia MADS box gene *FBP7* (*floral binding protein 7*) was found to prominently increase the auxin level at the ovule epidermis and significantly enhance the number of lint fibers, which ultimately increased the lint yield by more than 15% [39]. Various studies have indicated that auxin is primarily imported from other tissues in the ovules and transported into fiber cells through PIN-FORMED (GhPIN)-mediated auxin polar transport, rather than in situ synthesis [40,41,42]. Ovule-specific down-regulation of auxin efflux carrier (*GhPIN*) genes inhibited fiber initiation [40]. Furthermore, ectopic expression of cotton *GhPIN1a_Dt*, *GhPIN6_At*, and *GhPIN8_At* could enhance the length and density of trichomes on *Arabidopsis* leaves [41,42], implying that PIN proteins could contribute to fiber initiation. In addition to biosynthesis and transportation pathways, the auxin-signaling pathway also plays a role during cotton fiber development. Ectopic expression of cotton auxin response factors *GhARF2* and *GhARF18* could increase trichome initiations in *Arabidopsis*. These genes could be key factors in fiber initiation through the regulation of several transcription factors, including MYB and basic helix–loop–helix (bHLH) genes [43]. A recent study indicated that overexpression of *GhARF2b* using a fiber-specific promoter could increase fiber initiation, whereas the down-regulated expression of *GhARF2b* resulted in fewer fibers [44]. Compared with *GhARF2* and *GhARF18*, *GhIAA16* may have a negative effect during fiber initiation. Compared to the wild-type, *GhIAA16* transcripts showed higher expression immediately after flowering in a fiberless mutant [45]. Collectively, these results demonstrate the importance of auxin for fiber initiation.

### 4.2. GA

GA is a biologically active diterpene hormone that participates in diverse plant developmental processes, including seed germination, root and stem elongation, trichome development, and fruit ripening [46]. Overexpression of the cotton GA 20-oxidase homolog *GhGA20ox1* could promote fiber initiation by regulating gibberellin synthesis [47]. DELLA protein is a negative regulatory factor in the GA-signaling pathway; through interplay with TFs or important regulators, it represses binding to target genes or transcriptional activation [48]. Cotton DELLA protein GhGAI1 shows higher expression in XZ142FLM than its wild-type counterpart, implying a negative regulatory role in fiber initiation [49]. The GID1 protein can specifically bind to biologically active GA and further form a complex with DELLA protein. The cotton *GID1* homologous gene *GhGID1-1* is predominantly expressed in ovules, and it is inhibited by GA, implying a negative role in fiber initiation [50].

### 4.3. JA

JA also plays crucial roles in regulating plant growth, including growth inhibition, trichome development, and senescence [51]. Cotton ovules grown in medium with 0.001 μM JA exhibited increased fiber cell initiation; however, a higher concentration (2.5 μM) inhibited fiber initiation, implying that the appropriate concentration of JA is important for fiber initiation [52,53]. Real-time quantitative PCR (RT-qPCR) analysis of different fiber mutants indicated that four JA biosynthesis enzymes, the allene oxide cyclase (AOC) genes (*GhAOC1*-*GhAOC4*), were expressed at higher levels in −1 DPA ovules in fiberless mutants than in linted–fuzzless and linted–fuzzed lines. The expression of these genes increased under JA treatment. Meanwhile, these AOC genes were predominant expressed in −3–1 DPA ovules, particularly at −1 DPA, indicating their important regulatory role during fiber initiation. Taken together, the appropriate upregulation of *AOC* expression may be crucial during fiber initiation, while excessive production of *AOCs* might affect normal fiber development, suggesting a dose-dependent effect on fiber initiation [54]. This is consistent with the effects of different concentrations of JA in vitro ovule cultures [52,53]. The JASMONATE ZIM-DOMAIN (JAZ) protein is one of the crucial inhibitory factors in the JA-signaling pathway. *GhJAZ2* shows higher expression during the fiber initiation stage, and upregulated of *GhJAZ2* expression in cotton could inhibit fiber initials [53]. We believe that the effect of JA during fiber initiation is regulated in a dose-dependent manner.

### 4.4. BR

BRs are a kind of steroid hormones in plants and play important roles during cotton fiber development. In cotton, the application of the BR biosynthesis inhibitor, brassinazole (Brz) to floral buds completely blocked fiber differentiation, indicating that BRs play crucial roles during fiber initiation [55]. *GhDET2* encodes a steroid 5α-reductase that is highly expressed during the fiber initiation stage. The silencing of cotton *GhDET2* inhibited fiber initiation, while upregulation of *GhDET2* expression driven by the seed coat-specific promoter *pFBP7* enhanced fiber numbers [56]. The components of the BR-signaling pathway have been found to function during fiber development. 14-3-3 proteins play vital roles in the BR-signaling pathway. Suppressing the expression of cotton *Gh14-3-3L* significantly decreased the initiation of fiber cells, and a further analysis showed that Gh14-3-3L could interact with GhBZR1 to regulate fiber development [57]. Collectively, BRs mainly contribute to fiber initiation.

### 4.5. Cytokinin

Cytokinin plays a crucial role in plants, including in cell division, apical dominance, and the senescence of plant tissues and organs [58]. Supplementation of cotton ovule culture medium with exogenous cytokinin promotes the growth of ovules which imply that cytokinin plays an important role in ovule development [59]. Cytokinin oxidase/dehydrogenase (CKX) catalyzes the cleavage of the unsaturated side chain of cytokinin N6 and results in the loss of cytokinin activity, which is an important negative regulatory factor in cytokinin metabolism [60]. In conclusion, we suggest that cytokinin is required for fiber cell initiation in cotton.

### 4.6. ABA

ABA primarily functions in seed dormancy and stress responses [61]. In vitro application of ABA inhibited the initiation of cotton fibers, and this inhibitory effect was highly correlated with ABA levels [34]. Moreover, ABA also shows higher levels during the early stages of fiber formation in XZ142FLM lines [62]. All this evidence indicates that ABA could inhibit fiber cell initiation.

Moreover, there may be crosstalk between different phytohormones in the regulation of cotton fiber initiation. The cotton DELLA protein GhGAI1 inhibits fiber initiations, but adding IAA and BR greatly reduced the expression level of GhGAI1 at 0 DPA [49], indicating that auxin and BR may promote fiber initiation by repressing the DELLA protein GhGAI1, which connects auxin-, GA-, and BR-signaling during cotton fiber initiation (Figure 2).

**Table 2 plants-12-03771-t002:** Key phytohormone-related genes in regulating cotton fiber initiation.

Gene Type	Gene	Gene Function	References
Auxin-related	*iaaM*	Positive regulator of fiber initiation	[39]
	*GhPIN3a*	Positive regulator of fiber initiation	[40]
	*GhPIN1a_Dt*	Possible positive regulator of fiber initiation	[41,42]
	*GhPIN6_At*	Possible positive regulator of fiber initiation	[41,42]
	*GhPIN8_At*	Possible positive regulator of fiber initiation	[41,42]
	*GhARF2*	Possible positive regulator of fiber initiation	[43]
	*GhARF18*	Possible positive regulator of fiber initiation	[43]
	*GhARF2b*	Positive regulator of fiber initiation	[44]
	*GhIAA16*	Negative regulator of fiber initiation	[45]
GA-related	*GhGA20ox1*	Positive regulator of fiber initiation	[47]
	*GhGAI1*	Negative regulator of fiber initiation	[49]
	*GhGID1-1*	Negative regulator of fiber initiation	[50]
JA-related	*GhJAZ2*	Negative regulator of fiber initiation	[53]
	*GhAOC1-4*	Dosage effect on fiber initiation	[54]
BR-related	*GhDET2*	Positive regulator of fiber initiation	[56]
	*Gh14-3-3L*	Positive regulator of fiber initiation	[57]
CK-related	*GhCKXs*	Negative regulator of fiber initiation	[60]

## 5. Transcription Factors Involved in Fiber Initiation

Transcription factors play important roles in plant growth and development by regulating the transcription rate of downstream genes. In recent years, an increasing number of TFs have been reported to be involved in cotton fiber initiation [8,9,63,64,65] (Table 3).

### 5.1. MYB Transcription Factors

Over 400 *MYB* genes are preferentially expressed in at least one stage of fiber development. Suppression of the expression of *GhMYB109* leads to a partial (~8%) reduction in fiber initials, indicating that it plays a role in fiber initiation [66]. *GhCPC* is an R3-type MYB transcription factor; down-regulation of *GhCPC* expression in cotton leads to a reduction in fiber initiation [67]. Knock down of *GhMYB212* expression leads to shorter fibers and a lower lint index [68]. Recent evidence has indicated that genetic variations in *MYB5_A12* are associated with fiber initiation in tetraploid cotton [69]. The *MIXTA* factors are part of subgroup 9 (SBG9) from the R2R3 MYB family. Ten pairs of *MIXTA* (*GhMMLs*) homologs were isolated and identified in allotetraploid cotton [2]. The *N_1_N_1_* loci was identified as a defective allele of the At-subgenome homoeology of *GhMYB25*-like_*At* or *GhMML3_A12* that generates siRNAs from its 3′ end because the production of the overlapping antisense transcript post-transcriptionally silences both homoeologs of this gene [15]. Recently, we confirmed that the lint fiber factor GhMML4_D12 interacts with the WD40-repeat protein GhWDR in regulating cotton fiber development [33,70]. The combined form between them is similar but different to the MYB-bHLH-WD40 (MBW) complex in its regulation of trichome development, implying that cotton probably evolved a unique and independent process in the regulation of fiber initiation [64]. This finding also improved our understanding of the important function of *MML* genes in plants and in the regulation of cotton fiber production [64]. It is interesting that *GhMML4_D12* is arranged in tandem with *GhMML3*. These two closely related *MIXTA* genes regulate fiber initiation in two specialized cell forms (lint and fuzz fibers) [15,33,64]. *GhMYB25* (*GhMML7*)-silencing transgenic lines delay fiber initiation, but their overexpression leads to increased fiber initiation [71]. The reduced expression of *GhMYB109* in *GhMYB25*-silencing lines suggests that *GhMYB109* most likely acts downstream of *GhMYB25* [71]. Moreover, L1 box protein GbML1 regulates fiber development through interaction with GbMYB25 [72]. *GhMYB25*-like plays a critical role during fiber initiation, as the knock down expression of *GhMYB25*-like results in fuzzlessness, with fewer lint fibers on cotton seeds, similarly to the *N_1_NSM* mutant [73]. *GhMYB25*-like seems to be upstream of the other fiber-related MYB transcription factors. The expression of *GhMYB25* transcripts was almost abolished in *GhMYB25*-like-silenced lines, and *GhMYB109* also showed significant reduction. The siRNAs produced by *GhMML3_A12* in *N_1_NSM* plants may be involved in the gene silencing of MYB factors, including *GhMYB109*, *GhMYB25*, and *GhMML4* [15], which certainly confirms that *GhMYB25*-like acts upstream of these genes. Collectively, MYB genes, especially *MIXTA* factors, play important roles during cotton fiber initiation.

### 5.2. Homeodomain Leucine Zipper (HD-ZIP) Transcription Factors

HD-ZIP factors primarily function in regulating plant growth and development [74]. In cotton, one HD-ZIP transcription factor, *GhHD-1*, was identified; silencing this gene in cotton delayed the initiation of fibers, while upregulating the expression of *GhHD-1* enhanced the number of fiber initials [75]. *GhHD-1* transcripts were significantly decreased in *GhMYB25*-like-silenced lines, implying that *GhHD-1* acts downstream of *GhMYB25*-like [75]. A further analysis in transgenic lines showed that the expression of *GhHD-1* was not correlated with *GhMYB109* and *GhMYB25*, indicating that these genes may act in independent pathways to regulate fiber initiation. Cotton PROTODERMAL FACTOR1 gene (*GbPDF1*)-silenced transgenic lines had delayed fiber initiation [76], and *GhHD-1* may be directly involved in the regulation of *GbPDF1* through the core cis-element HDZIP2ATATHB2 [75].

### 5.3. TCP Transcription Factors

TCP proteins belong to a class of plant-specific transcription factors that play important roles in many biological processes in plants. Down-regulation of the expression of the TCP transcription factor *GbTCP* reduced the lint percentage [77]. *GhTCP14* is mainly expressed in fiber cells, especially during the initiation and elongation stages, and overexpression of *GhTCP14* in *Arabidopsis* enhanced the initiation of trichomes and root hairs. A further analysis indicated that *GhTCP14* binds to the promoters of *PIN2*, *IAA3*, and *AUX1*, implying that *GhTCP14* may act as a key regulator in the auxin-mediated differentiation of cotton fiber cell [78]. Recently, Wang discovered that *GhTCP14* is a kind of clock-regulated factor that targets genes crucial for translation and mitochondrial energy production, such as *RPL6* and *ATP5F1D*, indicating that fiber-related TFs are associated with clock-controlled genes [79].

### 5.4. WRKY Transcription Factors

Plant-specific WRKY proteins regulate various developmental and physiological processes. A recent study showed that *GhWRKY16* could contribute to cotton fiber initiation and elongation by directly binding to the promoter of *GhMYB25*, *GhMYB109*, *GhHOX3*, and *GhCesA6D-D11*. Moreover, GhWRKY16 is phosphorylated via the mitogen-activated protein kinase GhMPK3-1; and phosphorylated GhWRKY16 further promotes the transcription of downstream genes [80].

We found crosstalk between the phytohormone signaling pathways and transcription factors (Figure 3). The JAZ protein GhJAZ2 negatively regulates fiber initiation by interacting with GhMYB25-like [53], which connects JA signaling with the MIXTA factor. Solexa sequencing and Affymetrix gene chip analysis indicated that *GbTCP* has a positive regulatory effect on JA levels [77]. Moreover, *GhTCP14* binds to the promoters of auxin-related genes (*PIN2*, *IAA3*, and *AUX1*) to regulate cotton fiber initiation [78]. These findings demonstrate the relationship between *TCP* TFs and the JA and the auxin-signaling pathways (Figure 3).

**Table 3 plants-12-03771-t003:** Key transcription factors in regulating cotton fiber initiation.

Gene Type	Gene	Gene Function	Reference
MYB	*GhMYB109*	Positive regulator of fiber initiation	[66]
	*GhCPC*	Negative regulator of fiber initiation	[67]
	*GhMYB212*	Positive regulator of fiber initiation	[68]
	*MYB5_A12*	Positive regulator of fiber initiation	[69]
	*GhMML3_A12*(*GhMYB25*-like)	Positive regulator of fiber initiation	[15,73]
	*GhMML4_D12*	Positive regulator of fiber initiation	[33,70]
	*GhMYB25* (*GhMML7*)	Positive regulator of fiber initiation	[71]
HD-ZIP	*GbML1*	Possible positive regulator of fiber initiation	[72]
	*GhHD-1*	Positive regulator of fiber initiation	[75]
	*GbPDF1*	Positive regulator of fiber initiation	[76]
TCP	*GbTCP*	Positive regulator of fiber initiation	[77]
	*GhTCP14*	Positive regulator of fiber initiation	[78,79]
WRKY	*GhMPK3-1*	Positive regulator of fiber initiation	[80]

## 6. Sugar Signaling for Fiber Initiation

Developing cotton fibers are highly active sink cells that require sucrose (Suc) for differentiation, rapid expansion, and cellulose synthesis [81]. Cotton fibers need carbon from Suc, which is unloaded from the phloem of the ovule integument or seed coat. Before use, Suc should be split into either UDP-glucose (UDPG) and fructose (Fru) via sucrose synthase (Sus, EC 2.4.1.13) or hydrolyzed into glucose (Glc) and Fru via invertase (INV, EC 3.2.1.26) [81]. Compared to wild-type, fiber mutants (XZ142FLM and Li_1_) had increased Sus activity during the fiber cell initiation stage, implying that Sus probably acts as an early indicator for fiber initiation [82]. Sucrose transport proteins (SUTs) are important regulators of carbon allocation in plants [81]. Overexpression of the fungal SUT gene (*UmSrt1*) in the ovule epidermis and fibers enhanced the content of sugar, leading to an increase in fiber initials in cotton [83]. A vacuolar invertase gene, *GhVIN1*, shows high expression during the fiber initiation stage. Silencing *GhVIN1* resulted in a significant reduction in VIN activity and a completely fiberless phenotype [84]. A further analysis showed that *GhMYB25*-like might activate the transcription of VIN genes by binding to their promoters. VIN hydrolyzes Suc into glucose and fructose, thus producing hexose signals to activate the expression of *GhMYB25* and other MYB-related genes. Recently, co-expression networks generated by a comparative transcriptome analysis between *GhVIN1* RNA interference (RNAi) lines and XZ142FLM revealed common differentially expressed genes (DEGs) that regulate cotton fiber initiation, including *GhVIN1*, *GhMYB25*-like, *GhMYB25*, *GhMYB109*, *GhPDF1*, and *GhHD1* [85]. A further analysis indicated that *GhVIN1* and *GhMYB25*-like may participate in similar pathways that activating several transcription factors and plant hormone-related genes to regulate fiber initiation [85]. In addition, hexose-signaling may control the expression of genes associated with auxin biosynthesis, perception, and signal transduction [63,84]. Moreover, the fiber initiation-related factor *GhMYB212* binds the promoter of the sucrose transporter gene *GhSWEET12* to activate its expression; GhSWEET12 proteins transport sucrose into fiber cells. In fiber cells, GhVIN1 and GhSus catalyze sucrose to UDP-glucose or other sugars [68]. Collectively, the sugar-signaling pathway works together with MYB transcription factors and auxin-signaling to regulate cotton fiber initiation (Figure 3).

## 7. Small Signaling Molecules for Fiber Initiation

Ionic calcium (Ca^2+^) is a ubiquitous intracellular second messenger in plants and Ca^2+^ signaling plays an important role in the plant developmental processes [86]. Staining of cellular Ca^2+^ in cotton revealed that fiber cells accumulated more Ca^2+^ than other ovule cells, suggesting a function of Ca^2+^ in fiber development [87]. The expression of YC3.60 fluorescent Ca^2+^ marker in transgenic lines demonstrated the cellular and intracellular distribution of Ca^2+^ in cotton ovule epidermis and fiber cells [88].

Reactive oxygen species (ROS), including superoxide radical, hydrogen peroxide (H_2_O_2_), and hydroxyl radical, were found to regulate cell expansion in plants [89]. Using in vivo and in vitro cultured ovules, it was shown that 30% H_2_O_2_ could inhibit the retardation of fiber initials in Xinxiangxiaoji linted–fuzzless mutants (XinFLM) [90]. Pang demonstrated that the accumulation of ROS in ovule epidermal cells is essential for cotton fiber initiation [91]. The fiber-specific accumulation of Ca^2+^ during the initiation stage is similar to that of ROS in the ovule epidermis, suggesting a possible connection between these two signaling pathways in regulating cotton fiber initiation. The suppression of the calcium sensor *GhCaM7* delayed fiber initiation, and a further analysis indicated that *GhCaM7* may regulate the accumulation of ROS and serve as a molecular link between Ca^2+^ and the ROS-signaling pathways during early fiber development [92] (Figure 3). Recent evidence revealed that *GaHD1* promotes trichome and cotton fiber initiation through cellular H_2_O_2_ and Ca^2+^ signals [93]. These findings provide evidence that HD-ZIP transcription factors participate in cotton fiber initiation by regulating H_2_O_2_ and Ca^2+^ molecules (Figure 3).

## 8. Non-Coding RNAs (ncRNAs) and Histone Modification for Fiber Initiation

MicroRNAs (miRNAs) are non-coding small RNAs that are 18–24 nucleotides in length and play important roles during fiber initiation. A comparative miRNA omics analysis isolated seven cotton fiber initiation-related miRNAs in developing ovules and verified, through experiments, that the targets of these miRNAs participate in different metabolic processes and cellular responses, including transcriptional regulation, auxin and GA signal transduction, lignin biosynthesis, and actin bundles [94]. miR828 and miR858 regulate the function of homoeologous *MYB2* in both *Arabidopsis* trichome and cotton fiber development [95]. miR828 preferentially digests the transcripts of *GhMYB2_Dt* to generate trans-acting siRNAs (ta-siRNAs) [95]. These fiber-related genes can employ the derived siRNAs to target other downstream genes in the coordination of fiber cell development. A differential analysis of microRNAs between XZ142FLM and wild-type plants during fiber initiation identified 26 fiber initiation-related miRNAs, which target transcription factor-coding genes such as MYB, ARF, and LRR [96]. Recently, constitutive expression of the negative auxin-signaling regulatory Gh-miR167 (35S-MIM167) in cotton resulted in irregular fiber formation during the fiber initiation stage. A further analysis showed that, as the expression of *ARF6* and *ARF8* increased, the appearance of fiber initials on the surface of Gh-miR167 diminution lines was reduced [97], which suggests a coordination between plant hormone and regulatory miRNAs during cotton fiber initiation.

Long non-coding RNAs (lncRNAs) are transcripts with a length of at least 200 bp, which possess no significant coding ability and have been reported to contribute to fiber initiation. Wang found that most lncRNAs showed preferential expression in wild-type cotton ovules during the fiber initiation stage [98]. Specifically, the lncRNA LINC02 had significantly higher transcription levels in wild-type lines compared to fiberless mutants, implying that this gene may contribute to fiber initiation [98]. Silencing two lncRNAs (XLOC_545639 and XLOC_039050) in XZ142FLM via a virus-induced gene-silencing (VIGS) system increased the number of fiber initials on the ovules. This was the first functional identification of lncRNAs involved in fiber initiation, providing a basis for a deeper understanding of lncRNA functions during cotton fiber development [99]. Recently, 3288 lncRNAs were identified via high-throughput sequencing in a novel glabrous mutant ZM24*fl*, among them, a key lncRNA, MSTRG 2723.1, was isolated, which may activate the transcription of genes involved in pectin metabolism, the *GhMYB25*-mediating pathway, and the fatty acid metabolism to regulate fiber initiation in the ZM24 cultivar [100]. Taken together, the characterization and expression analysis of ncRNAs will contribute to future investigations of their roles in cotton fiber development.

Histone modification regulates gene expression in eukaryotes. The cotton histone deacetylase 5 gene *GhHDA5* is prominently expressed during fiber initiation; downregulated expression of *GhHDA5* suppressed fiber initiation and lint yield. Moreover, alterations in ROS homeostasis and increased autophagic cell death were found in 0 DPA ovules of the *GhHDA5* RNAi lines [101]. A further analysis indicated that the expression of fiber initiation-related factors *GhMYB25*-like and *GhHD1* was dramatically reduced in *GhHDA5* RNAi lines compared to the control [101]. Moreover, the addition of an exogenous histone deacetylation inhibitor (trichostatin A (TSA)) inhibited fiber initiation in in vitro culture of ovules. Further studies revealed that histone deacetylation could regulate some important phytohormone-related genes, thereby activating auxin, GA, and JA signaling pathways, while repressing ABA synthesis and signaling to promote fiber cell initiation [102]. These results indicate that there is crosstalk between histone deacetylation, with key transcription factors, and phytohormone pathways in the regulation of cotton fiber initiation.

## 9. Other Functional Genes for Fiber Initiation

Arabinogalactan proteins (AGPs) regulate many aspects of development in plants. The suppression of the fasciclin-like AGP GhFLA1 in cotton led to the retardation of fiber initiation and elongation [103]. A further analysis suggested that *GhFLA1* may play a role in cotton fiber development by regulating the composition of AGPs and the integrity of the primary cell wall matrix. Sphingolipids are bioactive molecules and crucial components of biomembranes that play roles in various biological processes, including plant growth, developmental regulation, and responses to stimuli. A comparative metabolomics analysis between XZ142FLM, wild-type, and Xinxiangxiaoji lintless–fuzzless mutant (Xinfl) plants revealed that sphingolipids and sterols play important roles during cotton fiber initiation [104]. Moreover, overexpressing of the cotton ceramide synthase gene *GhCS1* could inhibited fiber initiation and elongation by promoting the synthesis of ceramides, including dihydroxy long-chain bases and very-long-chain fatty acids [105]. The cytoskeleton-related profilin gene family have undergone strong purifying selection during cotton domestication. The over expression of the fiber-specific profilin gene *GhPRF1* could promote fiber cell initials; counter to no fiber cell initiation in RNAi lines, deep studies of the RNAi lines showed that auxin- and JA-signaling genes, together with sugar metabolism genes, were consistent with respect to the reduction in *GhPRF1* transcript levels [106], which further connect profilin protein with the phytohormone-signaling pathway and sugar metabolism in regulation of cotton fiber initiation (Figure 3).

## 10. Conclusions and Future Perspectives

Along with more high-quality genomes in cotton species being completed and the usage of mature gene operation technologies and methods of omics analysis, complicated regulatory networks and metabolic pathways will be revealed to offer new functional genes for cotton breeders in the improvement of crops. Understanding the factors that contribute to fiber initiation helps us to uncover the mechanisms and improve the yield potential of fiber. Here, we need a systematic and comprehensive genetic identification and regulatory analysis of functional genes related to cotton fiber initiation, which will broaden the possibilities for fundamental research in cotton biology.

In recent decades, several important genes associated with cotton fiber initiation have been identified and described. In this review, we summarized the different kinds of factors that contribute to this process and the crosstalk between them. We believe that, in regulating fiber initiation, different factors orchestrate together to form a network. MYB factors, especially the *MIXTA*-like gene *GhMML3* (*GhMYB25*-like), appear to be the most upstream transcription factors in the regulation of fiber initiation (Figure 3). Phytohormones, including auxin, GA, JA, BR and CK, can increase the amount of fiber initiation by regulating specific transcription factors (Figure 2). Of course, different hormone-signaling pathways and transcription factors can communication with each other. The cotton JAZ protein GhJAZ2 interacts with GhMYB25-like to control fiber initiation, indicating a direct role for JA in fiber initiation. IAA and BR can inhibit the expression of the cotton DELLA protein GhGAI1, which connects the auxin-, BR-, and the GA-signaling pathway (Figure 2). TCP transcription factors (*GbTCP* and *GhTCP14*) function during fiber initiation by regulating the level of JA or the transcription level of auxin-related genes (Figure 3)., The expression of cotton homologs associated with *MIXTA*, *MYB5*, *GL2*, and eight genes in the auxin, BR, GA, and ethylene pathways is activated during fiber development but this was inhibited in *N_1_NSM* [10], implying a synergistic effect between transcription factors and phytohormones in the regulation of fiber initiation. Moreover, sugar-signaling pathways play important roles in cotton fiber initiation. Knock down of *GhVIN1* resulted in fiberless seeds [84], similarly to *GhMYB25*-like-silenced transgenic lines [73]. Evidence showed that *GhMYB25*-like may bind to the promoter of *GhVIN1* to enhance its transcriptional level [63]. A recent study showed that *GhMYB25*-like and *GhVIN1* may have similar functions during fiber initiation through the regulation of several transcription factors (such as *GhMYB25*, *GhMYB109*, *GhPDF1*, and *GhHD1*) and phytohormone-related genes (such as *GhGA20ox1* and *GhDET2*) [85]. However, the elaborate crosstalk between these two important genes remains to be further studied. Small signaling molecules also interacts with transcription factors in the regulation of cotton fiber initiation. Newly research has shown that *GaHD1* contributes to trichome and fiber initiation by regulating ROS and Ca^2+^ signals (Figure 3). Moreover, the phytohormone-related regulation pathways also interact with ncRNAs, histone modification genes, and functional genes during cotton fiber initiation (Figure 3). Decreased fiber initials on the surface of Gh-miR167 diminution ovules due to increased expression of *ARF6* and *ARF8* indicated that ncRNAs could control fiber initiation through the regulation of auxin response factors (Figure 3). Histone deacetylation could affect fiber initiation by controlling phytohormone-related genes (Figure 3). The cotton profilin gene *GhPRF1* could contribute to cotton fiber initiation by controlling the expression level of auxin and JA-related genes (Figure 3). Notably, cotton fiber initiation is a complicated process that requires comprehensive alterations in gene expression through developmental and physiological pathways. Among them, transcription factors, especially *MIXTA*-like genes; phytohormone signals, especially auxin with GA-related genes; and sugar metabolism play dominant roles in regulating cotton fiber initiation. The coordination of various functional genes and signaling pathways integratesmany endogenous and exogenous factors into the process.

With the rapid development of biotechnology, more functional genes and regulation pathways involved in fiber initiation will be identified. Recent technological advances in single-cell sequencing (scRNA-seq) provide new tools for understanding the process of fiber initiation. Wang revealed a fiber-specific circadian-regulated gene expression program during cotton fiber development using scRNA-seq [79]. This research suggests that manipulating the circadian clock in fiber cells could be used to enhance fiber yield. A comparative analysis of the scRNA-seq data between XZ142FLM and wild-type plants further demonstrated the key roles of *GhMYB25*-like in the regulation of fiber initiation [107]. Future work will discover more functional genes and regulation pathways involved in cotton fiber initiation. We believe that more interactions between different kinds of genes will be found to enrich the network.

The biological breeding technology promoted by molecular biology has broken through the limitations of traditional breeding, making crop breeding more precise and efficient. The application of modern biotechnology in breeding will inevitably accelerate the breeding speed, shorten the breeding period, improve the breeding outcomes and pave the way for cotton variety improvement. We believe these findings and summaries provided in this review, especially those on the important functional genes responsible for fiber initiation, combined with modern molecular biology technology will be helpful in improving the yield in future cotton breeding work.

## Figures and Tables

**Figure 1 plants-12-03771-f001:**
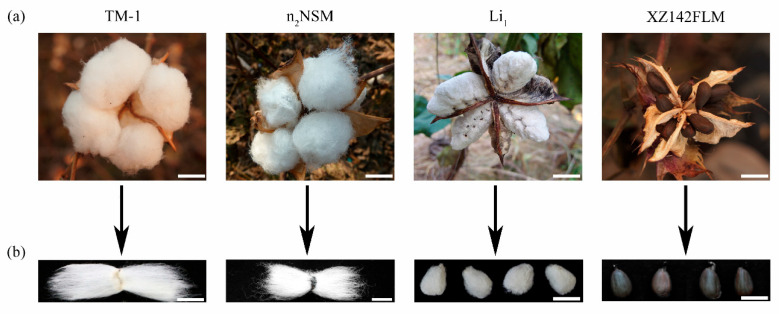
Cotton fiber phenotypes in cotton fiber in TM-1, n_2_NSM, Li_1_, and XZ142FLM. (**a**) Phenotypes of four representative varieties during the period of boll opening. Among them, white fibers with no black seeds are observed in TM-1, n_2_NSM, and Li_1_. Only black seeds are observed in XZ142FLM. Scale bars, 1.0 cm. (**b**) Examination of seed phenotype after fiber combing. TM-1 seeds have long lint and short fuzz fibers, n_2_NSM seeds only have long lint, Li_1_ seeds have short lint and fuzz fibers, and XZ142FLM seeds have no fibers attached. Scale bars, 1.0 cm.

**Figure 2 plants-12-03771-f002:**
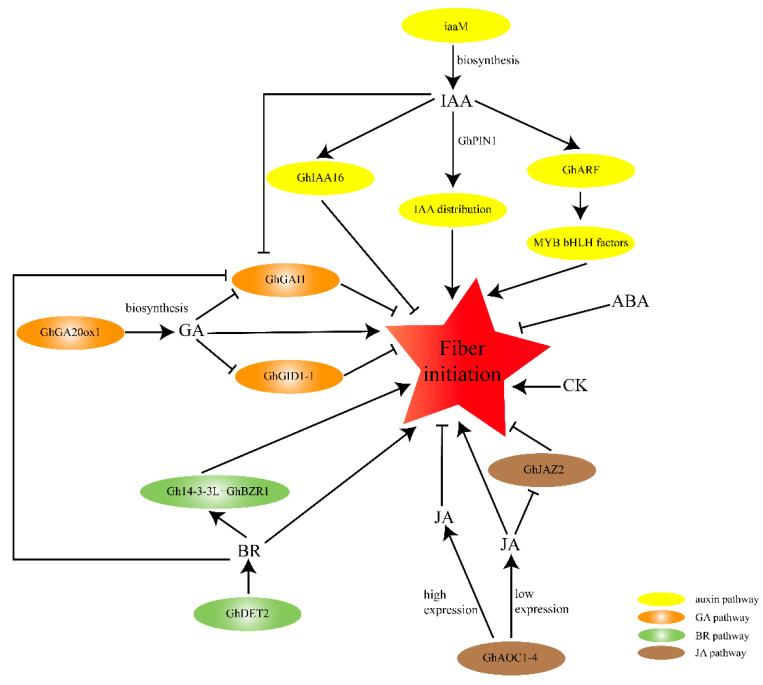
A schematic model of phytohormones of regulation of cotton fiber initiation by phytohormones. The arrows indicate the promotional effects and bars represent the inhibitory effects. Red five-pointed star represents fiber initiation; Yellow ovals indicate the auxin-signaling pathway; orange ovals represent the GA-signaling pathway; green ovals indicate the BR-signaling pathway; and brown ovals represent the JA-signaling pathway.

**Figure 3 plants-12-03771-f003:**
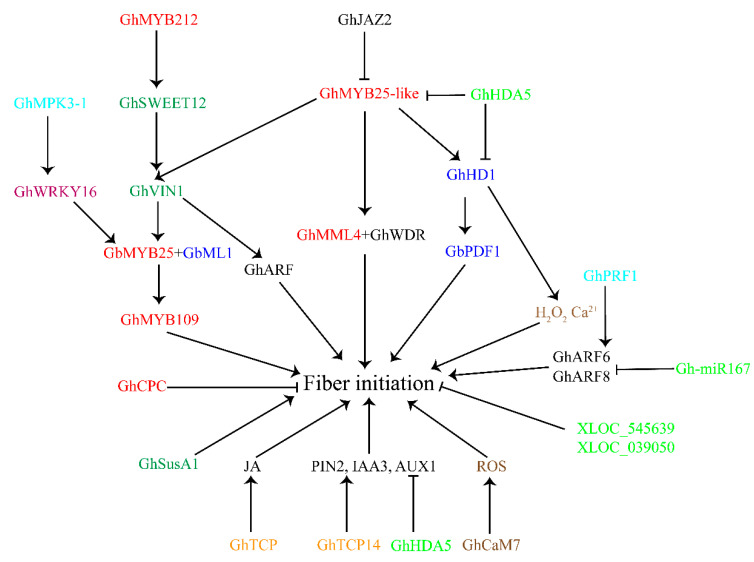
Diagram summarizing the roles of transcription factors, sugar signaling, small signaling molecules, and non-coding RNAs in cotton fiber initiation. The arrows indicate the promotional effects and bars represent the inhibitory effects. Red font represents MYB transcription factors; dark blue font indicates HD-ZIP transcription factors; purple font represents WRKY transcription factors; orange font indicates TCP transcription factors; dark green font represents sugar-signaling-related genes; brown font indicates small signaling molecules-related genes; light green font represents non-coding RNAs; light blue font indicates other functional genes.

## Data Availability

Not applicable.

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
