# Peer review of "Systematically and Comprehensively Understanding the Regulation of Cotton Fiber Initiation: A Review"

_plants, 2023, doi:10.3390/plants12213771_

Round 1

Reviewer 1 Report

Comments and Suggestions for Authors

Reviewer’s Comments

The submitted manuscript to Plants-MDPI entitled “Systematic and comprehensive understanding of cotton fiber 2 initiation: A review.

Although I could not find major shortcomings in this work but I would really like to see some metabolites which are responsible for the fiber initiation in cotton which can further polish this review.  

Author Response

Review1: Comments and Suggestions for Authors

Reviewer’s Comments

The submitted manuscript to Plants-MDPI entitled “Systematic and comprehensive understanding of cotton fiber 2 initiation: A review.”

Although I could not find major shortcomings in this work but I would really like to see some metabolites which are responsible for the fiber initiation in cotton which can further polish this review.  

Response: Thank you so much for the suggestion. Definitely, metabolites are responsible for cotton fiber initiation. Among them, sugar related signaling pathway make important roles during this process. In this review, we have specially summarized the sugar signaling for cotton fiber initiation. Lipid related signaling pathway mainly function in cotton fiber elongation, like cotton GPI-anchored lipid transfer protein, which regulates the transport of phosphatidylinositol monophosphates and cotton fiber elongation. In this review, we also summarized that over expression cotton ceramide synthase gene GhCS1 could inhibit fiber initiation and elongation by promoting the synthesis of ceramides containing dihydroxy long-chain bases and very-long-chain fatty acids. Moreover, to fulfill the function of metabolites for cotton fiber initiation, we have added ‘Comparative metabolomics analysis between XZ142FLM with its wild-type and Xinxiangxiaoji lintless-fuzzless mutant (Xinfl) revealed that sphingolipids and sterols make important roles during cotton fiber initiation [104]’ from line 456 to 458.

Reviewer 2 Report

Comments and Suggestions for Authors

In the paper titled “Systematic and comprehensive understanding of cotton fiber 2 initiation: A review”, the authors summarized the reported factors that effected fiber initiation by classifying them into categories including phytohormones, transcription factors, functional genes, non-coding RNAs, etc. Although this paper provides a systematic review of different factors in cotton fiber initiation, it is defective for publishing.

1)    Reviews on cotton fiber development, as well as fiber initiation, have been published a lot, this paper is lack of novelty. Despite some newly published papers in 2022 or 2023 were reviewed in this study, the majority of other contents, including tables and figures, was highly similar with other review papers (Wen et al., 2023. A comprehensive overview of cotton genomics, biotechnology and molecular biological studies; Fang et al., 2018. Unraveling cotton fiber development using fiber mutants in the post-genomic era; Tian and Zhang, 2021, MIXTAs and phytohormones orchestrate cotton fiber development.). 

2)    Line 82, I didn’t see any description about fuzz could reaches to 5-10mm in length in reference 11, and the most commonly recognized length about fuzz is less or approximately 5mm.

3)    Some of fiber initiation-related mutants in G. arboretum have also been reported recently, I suggest that they should also be summarized in Table 1.

4)    Line 241-244, the conclusion suggested by the authors is unsubstantiated. In the reference 57, the increase of fiber yield in caused by the increase of seeds numbers rather than the increase of fiber cell initiation in each seed. Since the readers of review paper like this study mostly are students, I do suggest that the authors don’t make any conclusion that is contrary to the fact. CK might play roles in regulating fiber cell initiation in cotton, but you could not get this conclusion form the present cited reference.

Author Response

Review2:

Comments and Suggestions for Authors

In the paper titled “Systematic and comprehensive understanding of cotton fiber 2 initiation: A review”, the authors summarized the reported factors that effected fiber initiation by classifying them into categories including phytohormones, transcription factors, functional genes, non-coding RNAs, etc. Although this paper provides a systematic review of different factors in cotton fiber initiation, it is defective for publishing.

  • Reviews on cotton fiber development, as well as fiber initiation, have been published a lot, this paper is lack of novelty. Despite some newly published papers in 2022 or 2023 were reviewed in this study, the majority of other contents, including tables and figures, was highly similar with other review papers (Wen et al., 2023. A comprehensive overview of cotton genomics, biotechnology and molecular biological studies; Fang et al., 2018. Unraveling cotton fiber development using fiber mutants in the post-genomic era; Tian and Zhang, 2021, MIXTAs and phytohormones orchestrate cotton fiber development.). 

Response: Thanks so much for the objective comments. Indeed, some studies with similar topic have been reported in recent years. As you have mentioned the three related reviews. In the paper of ‘Unraveling cotton fiber development using fiber mutants in the post-genomic era’. The author has summarized different kind of fiber mutants in Table 1 and displayed phenotype of these mutants in Figure 3. These results are similar with Table 1 and Figure 1 in our review. Obviously, they showed more detailed descriptions of fiber mutants. We feel so regretful to ignore this report during the period of writing this review. As you have suggested in the third question that we should report fiber initiation-related mutants in G. arboretum, we have added G. arboreum fuzzless mutant GA0149 and related research as supplement to describe fiber initiation-related mutants in our review. Moreover, we try to review a systematic and comprehensive understanding of cotton fiber initiation in this paper, fiber initiation-related mutants are extremely important resources to clarify the mechanism underlying fiber initiation. So, we believe the descriptions of cotton fiber initiation-related mutants could not be ignored.

In the paper of ‘MIXTAs and phytohormones orchestrate cotton fiber development’. We pay more attention to the MIXTA factors and phytohormone-related genes that responsible for cotton fiber initiation. In this review, we summarized different kind of factors that contribute to fiber initiation including plant phytohormones, transcription factors, sugar signal, small signal molecules, functional genes, non-coding RNAs and histone modification. We believe this review is more comprehensive. We show more detailed descriptions of plant phytohormones and transcription factors in this review. For example, we added many phytohormone-related genes (like GhIAA16, GhGAI1, GID1) and found crosstalk among IAA, GA, and BR in regulation of cotton fiber initiation. Moreover, we summarized nearly all different kind of transcription factors that are not limited to MIXTA factors. These transcription factors include MYB, HD-ZIP, TCP, and WRKY that are responsible for fiber initiation. In addition, we also systematically reviewed other kind of factors and summarized the crosstalk between them with the above phytohormones and transcription factors that function during fiber initiation. We believe all these findings help to enrich the network in the regulation of cotton fiber initiation.

In the paper of ‘A comprehensive overview of cotton genomics, biotechnology and molecular biological studies’. This paper reviewed nearly all aspects of studies in cotton biology and pointed out directions for future cotton research. Regulatory of fiber initiation have been clearly clarified in the paper, including transcription factors, involvement of phytohormones and sucrose-related genes. The important roles of transcription factors and phytohormones during fiber initiation are widely approved. To summarize the regulatory network of fiber initiation, we could not ignore this part. Whereas, this review just concentrates on the process of cotton fiber initiation and concludes most of the related studies, we believe this review is of great helpful for understanding the process of cotton fiber initiation systematically and comprehensively. Moreover, we classify different kind of the functional genes related to fiber initiation in this review, besides, we figure out the relationship between these factors to describe the elaborate regulatory networks underlying this process. We propose that all these factors coordinate with each other to regulate cotton fiber initiation. We believe further analysis in the interactions between these factors should be important research direction in the further work.

Above all, we believe the systematic and comprehensive review in the regulation of cotton fiber is the one novelty of this paper. In addition, further exploration the relationship between different kinds of factors is another novelty. We believe these findings and summaries in this review especially these important functional genes that responsible for fiber initiation combining with modern molecular biology technology will be helpful in improving yield in the future cotton breeding work.

  • Line 82, I didn’t see any description about fuzz could reaches to 5-10mm in length in reference 11, and the most commonly recognized length about fuzz is less or approximately 5mm.

Response: We have revised in the manuscript while fuzz fibers initiate on 4–5 DPA and ultimately reaches to approximately 5 mm in length’, moreover, we have replaced the related reference to ‘Johnson Kim J. From fuzz to fiber: identification of genes involved in cotton fiber elongation. Plant Physiol. 2020, 183, 23-24’ in this review.

  • Some of fiber initiation-related mutants in G. arboretum have also been reported recently, I suggest that they should also be summarized in Table 1.

Response: Sorry for the ignorance of the fiber initiation-related mutants in G. arboretum. We have added the related research in this reviewRecently, a ~6.2 kb insertion larlNDELFZ was found to be related with fuzzless seeds and decreased trichomes in G. Arboreum fuzzless mutant GA0149 by genome wide association study [23]. This insertion was proposed to enhance a dominant-repressor GaFZ from line 122 to 125.

4)    Line 241-244, the conclusion suggested by the authors is unsubstantiated. In the reference 57, the increase of fiber yield in caused by the increase of seeds numbers rather than the increase of fiber cell initiation in each seed. Since the readers of review paper like this study mostly are students, I do suggest that the authors don’t make any conclusion that is contrary to the fact. CK might play roles in regulating fiber cell initiation in cotton, but you could not get this conclusion form the present cited reference.

Response: Sorry for the invalid conclusion of the GhCKXs that contribute to cotton fiber initiation from reference 57. We have deleted the related description in this review. Thank you so much for your suggestion that we don’t make any conclusion that is contrary to the fact due to the readers of review paper like this study mostly are students. We have checked all the references and conclusion we proposed in this review carefully.

Reviewer 3 Report

Comments and Suggestions for Authors

1.     Is the Title appropriate to describe whole story of the research? Can be improved with the detailed and specific information.

2.     Does the Abstract represent the research? I think it will be better if you add more applications of this research.

3.     In Introduction, Is the content succinctly described and contextualized with respect to previous and present theoretical background and empirical research (if applicable) on the topic? Can be improved by adding more information about the previous studies related to the review topic.

4.     Is the Objectives of the research address correctly? Can be improved by using more specific words.

5.     How original is the topic? The topic is original but I saw many studies with similar topic, please add brief explanation about the unique of your review.

6.     Is the Result display in the correct way? Can be improved by adding the networking of the hormonal, transcription factors, and other functional genes in the cotton fiber initiation.

7.     Are the figures and tables clear and readable? Can be improved with higher resolution.

8.     Are the arguments and discussion of findings coherent, balanced and compelling? Can be improved with more coherent, balanced and compelling discussion.

9.     Are the conclusions consistent with the evidence and arguments presented? Yes.

10.  Are relevant data, citations, or references present? Yes.

Comments on the Quality of English Language

Please double check the Grammar error.

Author Response

Review3:

Comments and Suggestions for Authors

  1. Is the Title appropriate to describe whole story of the research? Can be improved with the detailed and specific information.

Response: To make the title more appropriate to describe whole story of the research, we have changed to ‘Systematic and comprehensive understanding the regulation of cotton fiber initiation: A review’.

  1. Does the Abstract represent the research? I think it will be better if you add more applications of this research.

Response: We have accepted your suggestion and add more applications in this research from line 20 to 22 ‘Our aim is to serve as a systematic and comprehensive review of different factors during fiber initiation that will provide the basics for further illustration of the mechanisms and offer theoretical guidance for improving fiber yield in the future molecular breeding work’.

  1. In Introduction, Is the content succinctly described and contextualized with respect to previous and present theoretical background and empirical research (if applicable) on the topic? Can be improved by adding more information about the previous studies related to the review topic.

Response: We have accepted your suggestion and add ‘Previous studies have shown that transcription factors (TFs) and phytohormone make important roles during cotton fiber development [7-9]’ from line 54 to 55.

  1. Is the Objectives of the research address correctly? Can be improved by using more specific words.

Response: We have accepted your suggestion and improved by using specific words.

  1. How original is the topic? The topic is original but I saw many studies with similar topic, please add brief explanation about the unique of your review.

Response: Indeed, some studies with similar topic have been reported in recent years. Whereas, this review just concentrates on the process of cotton fiber initiation and concludes most of the related studies, we believe this review is of great help for understanding the process of cotton fiber initiation systematically and comprehensively. Moreover, we classify different kinds of the functional genes related to fiber initiation in this review, besides, we figure out the relationship between these factors to describe the elaborate regulatory networks underlying this process. We propose that all these factors coordinate with each other to regulate cotton fiber initiation. We believe further analysis in the interactions between these factors should be important research direction in the further work.

Above all, we believe the systematic and comprehensive review in the regulation of cotton fiber is the one novelty of this paper. In addition, further exploration the relationship between different kinds of factors is another novelty.

  1. Is the Result display in the correct way? Can be improved by adding the networking of the hormonal, transcription factors, and other functional genes in the cotton fiber initiation.

Response: Thanks for the suggestion by adding the networking of the hormonal, transcription factors, and other functional genes in the cotton fiber initiation. Actually, figure 3 in this review have synthesized hormonal, transcription factors, and other functional genes together to form the network in regulation of fiber initiation. In figure 3, auxin-related genes (PIN2, IAA3, AUX1, GhARF), JA, and JA-related gene (GhJAZ2) have been combined into the network, moreover, the legend of figure 3 revealed sugar signaling, small signaling molecules, non-coding RNAs, and other kind of functional genes indicated different colour.

  1. Are the figures and tables clear and readable? Can be improved with higher resolution.

Response: We have replaced all figures with higher resolution to make this manuscript clear and readable.

  1. Are the arguments and discussion of findings coherent, balanced and compelling? Can be improved with more coherent, balanced and compelling discussion.

Response: We have rewrite part of the discussion to make it more coherent, balanced, and compelling.

  1. Are the conclusions consistent withthe evidence and arguments presented? Yes.

Response: Thanks so much for the approval.

10.  Are relevant data, citations, or references present? 

Response: Thanks so much for the approval.

Round 2

Reviewer 2 Report

Comments and Suggestions for Authors

The authors have revised the manuscript according to previous comments from reviewers, and I think this manuscript meets the demands of the current journal.